# Hemolysin-like Protein of ‘*Candidatus* Phytoplasma Mali’ Is an NTPase and Binds *Arabidopsis thaliana* Toc33

**DOI:** 10.3390/microorganisms13051150

**Published:** 2025-05-17

**Authors:** Kajohn Boonrod, Alisa Konnerth, Mario Braun, Gabi Krczal

**Affiliations:** RLP AgroScience GmbH, Institute for Plant Research, Breitenweg 71, 67435 Neustadt an der Weinstraße, Germany; alisa.strohmayer@web.de (A.K.); mario.braun@agroscience.rlp.de (M.B.); gabi.krczal@agroscience.rlp.de (G.K.)

**Keywords:** phytoplasma, hemolysis, NTPase, Toc33, NTP-binding protein

## Abstract

‘*Candidatus* Phytoplasma mali’ is associated with apple proliferation, a devastating disease in fruit production. Using genome analysis, a gene encoding a hemolysin-like protein was identified. It was postulated that this protein could be an effector. However, the function of this protein is unknown. It is shown that the hemolysin-like protein binds to a GTP binding protein, Toc33, of *Arabidopsis thaliana* in yeast two-hybrid analysis and that the Toc33-binding domain is located in the C-terminus of the domain of unknown function (DUF21) of the protein. The biochemical studies reveal that the protein can hydrolyze phosphate of purine and pyrimidine nucleotides. Transgenic *Nicotiana benthamiana* plants expressing the protein show no discernible change in phenotype. Phytoplasma have a much-reduced genome, lacking important genes for catabolic pathways or nucleotide production; therefore, the hemolysin-like protein plays a role in the uptake of plant nucleotides from their host and hydrolyzes these nucleotides for energy and their own biosynthesis.

## 1. Introduction

Phytoplasmas infect various plant species worldwide, causing dramatic yield losses in crop production [1]. Although phytoplasma genomes contain genes for basic cellular functions, they lack genes for amino acid biosynthesis, fatty acid biosynthesis, the tricarboxylic acid cycle, and oxidative phosphorylation and encode even fewer metabolic functional proteins. In addition, phytoplasma genes encode several proteins that have been identified as effectors [2]. Nevertheless, many proteins and their functions have not yet been identified. One of them is the hemolysin-like protein (HLP).

The phytoplasma HLPs are typically encoded by the ORFs *hlyC* (also called *hlyIII*), and *tlyC* and can be found in several phytoplasma, including ‘*Candidatus* Phytoplasma asteris’ strains AY-WB and OY-M [2,3], ‘Ca. P. australiense’ [4], ‘Ca. P. phoenicum’ [5] and ‘Ca. P. ziziphi’ [6], and Mulberry dwarf (MD) phytoplasma [7]. Moreover, they can also be found in other plant pathogenic bacteria like *Xylella* [8]. The function of phytoplasma HLPs has been discussed as playing a role in phytoplasma virulence [9].

The HlyC-group of phytoplasma HLPs (HlyC-HLP) have sequence similarities to a *Staphylococcus epidermis* hemolysin III protein which has a conserved domain similar to the hemolysin inner membrane protein, YqfA [3,4], while the TlyC-group (TlyC-HLP) is similar to the cluster of orthologous group (COG) 1253 of hemolysins and related proteins which contains cystathionine-beta-synthase (CBS) domains, generally found in two or four copies within a protein, and may play a regulatory role. However, its exact function is unknown [10,11].

‘Ca. P. mali’ is the causal agent with it the agent associated with apple proliferation disease (AP), one of the economically most important phytoplasma diseases in Europe, affecting fruit quality and productivity of infected trees. It mostly localizes in plant phloem and is mainly transmitted from plant to plant by the leaf suckers *Cacopsylla picta* and *Cacopsylla melanoneura* [12]. In the ‘Ca. P. mali’ genome, *hlyC* encodes a hemolysin III-related protein (CAP18505), and *tlyC* encodes the TlyC-HLP type (WP_012504424), which, interestingly, carries two additional domains, the N-terminal domain of unknown function (DUF21, pfam01595), a transmembrane region, and the C-terminal CorC-HlyC domain (pfam03471) (Figure 1B). Therefore, it is also referred to as the HlyC/CorC family transporter protein [9]. The same feature is also found in the TlyC-HLP of ‘Ca. P. asteris’ strains AY-WB and OY-M and ‘Ca. P. australiense’ [2,3,4].

Although the TlyC-HLP of the ‘Ca. P. mali’ phytoplasma is related to hemolysins, the presence of extra domains suggests that it is different from the identified hemolysin of other phytoplasma strains. Currently direct evidence of HLPs being pathogenicity factors is missing [4], although it has been proposed that phytoplasma HLP may be virulence factors acting as toxins or antimicrobial compounds [9], but there is no evidence to support this prediction. Therefore, in this work, we investigated the function of the TlyC-HLP from ‘Ca. P. mali’ strain PM19 (HLP_PM19_).

## 2. Materials and Methods

### 2.1. Origin of HLP_PM19_ DNA

HLP_PM19_ (GeneBank accession number MN207066) was amplified from DNA of ‘Ca. P. mali’ strain PM19 that was previously transmitted from field-collected *C. picta* to healthy test plants of *Malus x domestica* [12]. Total nucleic acids from plant tissue was extracted with a modified cetyltrimethylammonium bromide-based protocol [12]. Primers for gene amplification were chosen based on genomic data from the corresponding gene of ‘Ca. P. mali’ strain AT [9]: 5′-ATGGTTTTAATAATCTTC-3′ and 5′-TTAATTTTTGTCTGAAATTG-3′.

### 2.2. Yeast Two-Hybrid (Y2H) Screening

Library scale Y2H screening was performed using a Matchmaker Gold Yeast Two-Hybrid System (Takara Bio USA, Inc., Mountain View, CA, USA). For bait, an N-terminal deletion mutant of HLP_PM19_ (amino acids 134–428) was cloned into the pGBKT7 expression vector for binding domain (BD) fusion. For prey, the Mate&Plate^TM^ Library Universal Arabidopsis (normalized) (Takara Bio USA, San Jose, CA, USA) was used. It contains cDNA derived from 11 *A. thaliana* tissues, cloned into the pGADT7 vector (LEU2) for activation domain (AD) fusion. The Y2H screen was performed according to manufacturer’s protocol and as described in detail elsewhere [13]. Small-scale Y2H to test only one bait and one prey candidate against each other was performed using co-transformation and as described [13]. Different deletion mutants of HLP_PM19_ were fused to the BD for bait and AtToc33 to AD for prey.

### 2.3. Recombinant Expression and Purification of HLP_PM19_ and AtToc33

For HLP_PM19-∆TM_ expression, the HLP_PM19_ without amino acids 1 to 199 of HLP_PM19_ was amplified and cloned as N-terminal fusion to a thioredoxin (Trx) tag and C-terminal to a 6×His- and Strep-tag, into plasmid pET32a+ (Novagen, Merck KGaA, Darmstadt, Germany), resulting in the fusion protein Trx-HLP_PM19-∆TM His-Strep_.

For expression of AtToc33, a version of the gene was used missing the C-terminal transmembrane domain and containing an arginine (R) substituted by alanine (A) at position 130 (R130A mutant) [14]. This gene was amplified from the positive clone found in the Y2H screen, pGAD-AtToc33, and cloned into the expression vector pET28a+ (Novagen) fused to N- and C-terminal 6×His-tag. The introduction of the R130A mutation was conducted using the primers 5′-CGTTTGGATGTGTATGC-AGTCGATGAGCTAG-3′ and 5′-CTAGCTCATCGACTGCATACACATCCAAACG-3′ via site-directed mutagenesis [15]. All obtained recombinant plasmids were sent for sequencing at Starseq (Mainz, Germany).

For protein expression, a single colony of transformed BL21 (DE3)+ cells harboring the plasmid were grown at 37 °C overnight and sub-cultured 1:100 in 500 mL 2×TY broth and grown at 37 °C to an OD_600_ of 0.5. Expression of the target protein was induced with 1.0 mM isopropyl β-D-thiogalactopyranoside (IPTG, Iris Biotech GmbH, Marktredwitz, Germany) and incubation overnight at 14 °C. Cells were harvested by centrifugation, and protein was extracted from the pellet using 5 mL BugBuster^TM^ Protein Extraction Reagent (Millipore, Merck, Darmstadt, Germany).

For protein purification, Protino^®^ Ni-NTA Agarose (Macherey-Nagel GmbH & Co. KG, Düren, Germany) was added to the cleared lysate, and the mixture was incubated 1 hour (h) rotating at room temperature (rt). Agarose beads were washed with wash buffer containing 25 mM Tris-HCl, pH 7.5, 300 mM NaCl, 0.01% Triton X-100, and 5 mM Imidazole. Proteins were eluted with 150 µL elution buffer containing 25 mM Tris-HCl, pH 7.5, 150 mM NaCl, 0.01% Triton X-100, and 300 mM Imidazole. Protein was stored at 4 °C.

### 2.4. ELISA

For direct ELISA assay, 2 µg of purified AtToc33_ΔTM-R130A-His_ was coated on an ELISA microarray plate overnight at 4 °C. Wells were washed 3 times with PBS buffer and blocked using 3% BSA in PBS buffer for 2 h at rt. Wells were washed again, and 1 µg of Trx-HLP_PM19-His-Strep_, or Trx_His-Strep_ as a negative control, was added and incubated for 2 h at rt. Presence of protein binding was detected after thorough washing of the wells using Strep-Tactin-conjugated POD (IBA, Lifescience, Göttingen, Germany). The colorimetric reaction was started with tetramethylbenzidine (TMB) and stopped with sulfuric acid after 5 min. Yellow coloring was measured using an ELISA reader (Thermo electron, Muttiskan ascent, Thermo Fisher Scientific, Darmstadt, Germany) at 450 nm. The experiment was repeated at least 3 times.

### 2.5. GTP-Binding Assay

In vitro GTP-binding assay was performed according to a protocol published elsewhere [16] with some modifications. A quantity of 10 µM of purified Trx-HLP_PM19 His-Strep_, AtToc33_ΔTM-R130A-His_, Trx_His-Strep_, or BSA was incubated with 0.1 µM [α32P]GTP in 10 µL binding buffer (10 mM MgCl_2_, 0.1% Triton X-100, and 1 mM ATP in 20 mM Tris-HCl, pH 8.0, 100 mM NaCl, 1 mM DTT, and 1 mM EDTA for 1 h on ice. A nitrocellulose membrane was soaked with binding buffer and air-dried, and 2 µL of reaction mix was spotted on the membrane. The membrane was air-dried and washed 3 times with ice-cold washing buffer (5 mM MgCl_2_, 0.3% Tween-20 in 20 mM Tris-HCl, pH 8.0). Bound [α32P]GTP was detected using a phosphor imager (Biorad pharosFX plus molecular image, Bio-Rad Laboratories GmbH, Feldkirchen, Germany). The experiment was repeated at least 3 times.

### 2.6. Immuno-Fluorescence Histochemical Staining

The ‘Ca. P. mali’ strain AT-infected *N. occidentalis* was kindly provided by Kerstin Zikeli, Julius Kühn-Institut (JKI), Dossenheim, Germany). The purified recombinant HLP was sent for generating a polyclonal anti-HLP in rabbit ( Davids Biotechnologie GmbH, Regensburg Germany). Mounting and preparation of the sections was performed as described in [17] with the following modifications: xylene was substituted with HistoChoice clearing agent (Sigma). After dewaxing and rehydration of the sections, primary antibody binding of hemolysin was performed with 400 µL per slide of purified polyclonal anti-HLP rabbit serum diluted 1:100 in 1×PBS containing 3% BSA at rt overnight. The sections were carefully washed three times for 10 min each with 400 µL of 1×PBS containing 0.88% NaCl, 0.1% Tween-20, and 0.8% BSA (washing solution 1), and a final washing step was carried out for 1 h in 1×PBS containing 0.8% BSA (washing solution 2). Immunofluorescence staining of the antibody-decorated HLP was then performed by incubation with 200 µL of anti-rabbit antibody conjugated to Alexa 568 diluted 1:1000 in washing solution 2 for 3 h. Unbound secondary antibody was washed away with washing solution 1 for 15 min. To visualize immuno-fluorescence staining, a small drop of washing solution 2 was dropped on the treated samples, and images were captured using a Zeiss Observer Z1 inverse microscope with an attached LSM 510 confocal laser scanning system (Carl Zeiss, Oberkochen, Germany). The experiment was repeated more than 3 times with different plant samples.

### 2.7. Nucleotidase Activity Assay

The assay reaction was adapted from the one described [17,18]. The reaction consists of 30 µL assay buffer (40 mM Tris, 80 mM NaCl, 8 mM magnesium acetate, 1 mM EDTA, pH 7.5) with 10 µL of 10 mM NTP and 1 µg of enzyme sample (or assay buffer for the negative control). The assay was incubated for 1 h at rt and then 10 µL of solution 1 [0.5 g ammoniummolybdate pre-dissolved in 3.5 mL semi-concentrated nitric acid (63%), and the volume was adjusted to 10 mL with distilled water and vortexed. Thereafter, 10 µL of solution 2 [9 µL of 100 mM ascorbic acid in water and 1 µL of freshly prepared 5% Tin(II)chloride dissolved in 50% concentrated hydrochloric acid] were added into the assay reaction. After 5 min at rt, the developed blue color was measured with a spectrophotometer (Thermo electron, Muttiskan ascent, Thermo Fisher Scientific, Darmstadt, Germany) at 630 nm. The experiment was repeated at least 3 times.

### 2.8. Thin Layer Chromatography (TLC) Analysis

For TLC, the nucleosidase activity assays were performed as before, with the exception that 2 mM NTP was used in order to better see spots on the chromatogram. A quantity of 10 µL of nucleosidase activity assays were spotted on a Polygram CEL 300 PEI/UV254 sheet (Macherey-Nagel GmbH & Co. KG, Düren, Germany) and developed with 1.2 M LiCl for 1 h. To stain phosphate in di- and triphosphate nucleotides with molybdenum, a 5 min UVC exposure step (Biometra FLX20.M transilluminater, Vilber, Eberhardzell, Germany) was performed to break down NTPs. (Polyphosphates were already visible as dark spots under UV light by now, but the free phosphate did not quench fluorescence on the UV254 PEI sheet). The TLC sheet was then carefully placed in a large tray containing solution 1, by placing the edge of the sheet into the solution and then turning the sheet face down until the solution covered the entire sheet. Holding the sheet upright, excess solution was allowed to drip off (we did not use tissue paper during this process, as it may have contained a significant amount of free phosphate), and then the sheet was placed face-up on a plastic rack to dry. The TLC sheet was then immersed in the same fashion as described before in solution 2 (10 mL) and quickly removed when the entire surface was in contact with the solution. Free phosphate developed as dark blue or greenish spots on the chromatogram. The experiment was repeated at least 3 times.

## 3. Results

### 3.1. Sequence Analysis of HLP_PM19_

The HLP gene studied was derived from ‘Ca. P. mali’ strain PM19 [12], and is thus called HLP_PM19_ (GenBank accession number MN207066). The gene product of HLP_PM19_ is a 49.5 kDa protein and corresponds to a protein of the ‘Ca. P. mali’ strain AT that is described as an HlyC/CorC family transporter protein (Accession number WP_012504424) which was identified as TlyC type (COG1253). HLP_PM19_ and its AT strain homolog share an identity of over 99% on amino acid level with only three amino acids being substituted (Figure 1B). A phylogenetic tree comparing a selection of phytoplasmas HlyIII and TlyC proteins shows that both protein groups form an own cluster of highly conserved proteins (Figure 1A).

The protein structure prediction indicated that HLP_PM19_ contains an intracellular N-terminal region from amino acids 1 to 46, a transmembrane region from amino acids 47 to 133, with three transmembrane helices and an extracellular C-terminal region from amino acids 134 to 428 (using TMHMM prediction of transmembrane helices [19,20]. Moreover, it contains a CorC/HlyC-associated CBS pair (pfam00571) located within amino acids 199 to 316. Furthermore, amino acids 1 to 176 were identified as DUF21 (pfam01595) and amino acids 333 to 413 as CorC-HlyC (pfam03471), a transporter-associated domain of so far unknown function, that might be involved in modulating transport of ion substrates [9]. These TlyC domains of ‘Ca. P. mali’ have also been reported for ‘Ca. P. asteris’ strain AY-WB, strain OY-M, and ‘Ca. P. australiense’ [2,3,4,9]. Due to the high conservation amongst TlyC proteins, a similar domain structure can be suggested for TlyC of other phytoplasmas (Figure 1B).

**Figure 1 microorganisms-13-01150-f001:**
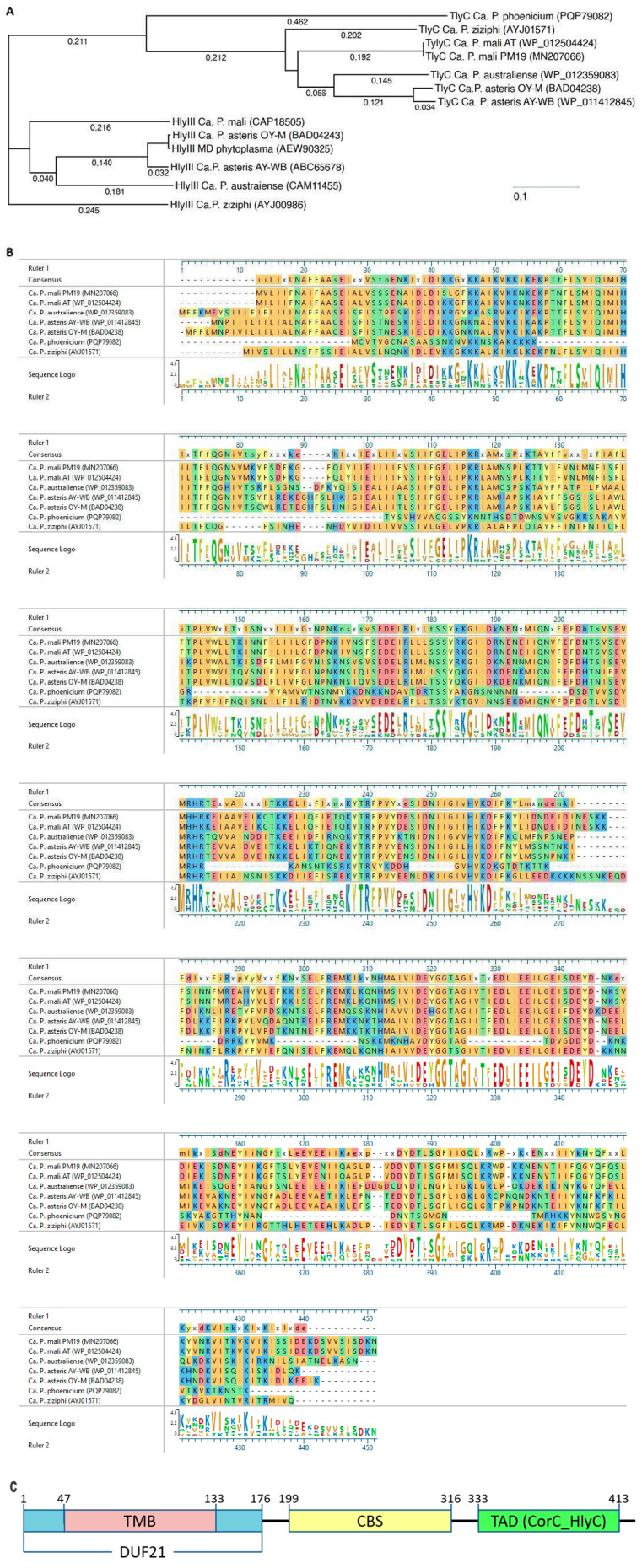
Comparison of hemolysin-like protein ‘Ca. P. mali’ strain PM19 with other related proteins. Codes in brackets represent NCBI accession numbers. (**A**) Phylogenetic tree of different phytoplasmas TlyC and HlyC proteins. (**B**) Amino acid sequences alignment of phytoplasma TlyC proteins. Degree of conservation is color-coded and out listed in sequence logo. (**C**) Protein domains of TlyC protein of ‘Ca. P. mali’ strain PM19 are drawn. Transmembrane region (TMB) with three transmembrane helices is amino acids 47 to 133; domain of unknown function DUF21 is amino acids 1 to 176; CBS containing region is amino acids 199 to 316; and transporter associated domain (TAD) CorC_HlyC is amino acids 333 to 413.

### 3.2. Recombinant Expression of HLP_PM19_ Does Not Change the Plant Phenotype

It has been proposed that the phytoplasma HLP could be an effector [3,9]. To prove this hypothesis, we produced transgenic *Nicotiana benthamiana* and *Arabidopsis thaliana* plants expressing full-length HLP_PM19_ and transmembrane-deleted HLP_PM19∆TM_ under 35S promoter control. The result showed that both transgenic plant species did not show any remarkable phenotypic changes (Figure 2). Since the phytoplasma colonize mostly the phloem, we also produced transgenic *N. benthamiana* and *A. thaliana* plants expressing the full-length and HLP_PM19∆TM_ under a phloem specific promoter (AtSUC2, [21]). Neither of these plants showed phenotypic changes.

### 3.3. HLP Is Located on the Phytoplasma Membrane

Since the HLP_PM19_ protein has two transmembrane domains, it was hypothesized that it might be expressed and localized at the phytoplasma membrane, like the immunodominant membrane proteins [22], and that it might bind some host proteins and could act as a regulatory protein for phytoplasma survival. To prove that HLP_PM19_ is expressed on the phytoplasma cell membrane, we performed an immuno-fluorescence histochemical staining using a specific antibody binding to HLP_PM19_. The result, in Figure 3, showed that HLP_PM19_ was specifically detected in the phloem of plants infected with the ‘Ca. P. mali’ strain AT (right panel), but not in the uninfected plant (control, left panel), suggesting that the HLPs are localized on the surface of phytoplasma cells.

### 3.4. HLP_PM19_ Interacts with AtToc33 in Y2H Assay

We further performed a Y2H screen to find interaction partners of HLP_PM19_ that may indicate a possible function of this protein. To avoid false results due to non-soluble protein expression caused by transmembrane helices, a truncated version of the protein missing amino acids 1 to 133 (ΔTM, Figure 4A), was used in all experiments.

In Y2H, HLP_PM19-ΔTM_ fused to the Gal4 binding domain (BD) was used for bait (expression plasmid pGBKT7-*HLP_PM19-ΔTM_*), and a normalized *A. thaliana* library (Takara Bio USA, Inc., Mountain View, CA, USA) fused to the Gal4 activation domain (AD) as prey (expression plasmid pGADT7). In total, 7.1 × 10^7^ clones containing both prey and bait were screened for interaction upon resistance to antibiotic aureobasidin A (AbA), expression of α-galactosidase, and growth on media lacking histidine and adenine. One clone was found to activate all four reporter genes. Plasmid isolation and sequencing showed that the prey plasmid of this clone contained the full-length gene of the *A. thaliana* translocase of chloroplast 33 (AtToc33, accession number O23680).

During transformation, yeast cells are able to take up more than one plasmid, which might lead to false positive results in library-scale Y2H screens. To confirm the interaction of HLP_PM19-ΔTM_ with AtToc33, a co-transformation was conducted on the bait and the prey plasmids, the latter isolated from the original positive transformant. For negative control, an empty pGBKT7 plasmid expressing only the BD was performed. The co-transformants were screened for resistance to AbA and expression of α-galactosidase. The result showed that combination of HLP_PM19-ΔTM_ with AtToc33 showed a clear positive signal (Figure 4B) whereas the negative control did not (Figure 4B). Thus, the result confirms the interaction of HLP_PM19 ΔTM_ with AtToc33 in the Y2H assay.

### 3.5. Amino Acids 133 to 198 of HLP_PM19_ Are Important for Interaction with AtToc33 in Y2H

To localize the AtToc33 binding site of HLP_PM19_, two deletion mutants were generated. HLP_PM19-N_, in which amino acids 134–316 were deleted from the transmembrane region to the end of the CBS-containing domain, and HLP_PM19-C_, in which amino acids 199–428 were deleted from the CBS-containing domain to the end of the gene (Figure 4A). Both genes were fused to the BD and tested against AtToc33-AD in Y2H analysis. The results in Figure 4B show that HLP_PM19-N_ showed a clear positive signal in Y2H, whereas HLP_PM19-C_ was negative), indicating that the binding domain is located in between amino acids 134 and 198. To test whether this region is indeed responding to the interaction, we generated a gene encoding only amino acids 134 to 198, containing mainly the C-terminal area of the DUF21 domain (HLP_PM19-DUF21-C_, Figure 4A), fused to the BD and performed a Y2H. The Y2H results (Figure 4B) clearly indicate that amino acids 133 to 198 of HLP_PM19_ are responsible for the interaction with AtToc33 in Y2H.

### 3.6. HLP_PM19-ΔTM_ Binds AtToc33 in ELISA

To confirm the binding of HLP_PM19-ΔTM_ to AtToc33 obtained from Y2H assay, we expressed the recombinant proteins in *E. coli* and tested their binding in ELISA. Expression of HLP_PM19-ΔTM_ alone did not yield enough soluble protein for purification thus, HLP_PM19-ΔTM_ was fused to Trx at its N terminus and 6×His for purification, and Strep-tag for detection was fused at the C-terminus, yielding the 57 kDa fusion protein Trx-HLP_PM19-ΔTM-His-Strep_. AtToc33 was expressed and purified as described by Wang and co-workers [14] using a truncated gene lacking the C-terminal transmembrane domain (ΔTM) and containing an alanine-substituted arginine at position 130 (R130A), resulting in AtToc33_ΔTM-R130A_. This construct was proved to be highly expressed in *E. coli*, and the purified protein was fully functional [14].

After protein expression in *E. coli* and purification using Ni-NTA resin, the purity of both proteins was determined by SDS-PAGE followed by coomassie brilliant blue staining (Figure 5A). The results showed that the protein samples of AtToc33_ΔTM-R130A-His_ and Trx-HLP_PM19-ΔTM-His-Strep_ showed high purity (Figure 5A); thus, they were further used for determining their binding affinities in an ELISA.

A direct ELISA assay was performed to confirm interaction of both proteins in vitro. The ELISA result showed that Trx-HLP_PM19-ΔTM-His-Strep_ can indeed bind AtToc33_ΔTM-R130A-His_ in vitro (Figure 5B). This result confirms the interaction of Trx-HLP_PM19-ΔTM-His-Strep_ with AtToc33 in Y2H analysis.

### 3.7. HLP_PM19_ Binds and Hydrolyzes GTP

The protein sequence analysis showed that HLP_PM19_ contains a CBS domain functioning as ATP/GTPase. In addition, the results show that HLP_PM19_ binds to AtToc33, which is part of a protein translocation complex in the outer envelope membrane of chloroplasts and binds precursor proteins [23,24]. Since Toc33 binds and hydrolyzes GTP in an in vitro assay [14], it was also interesting to investigate whether HLP_PM19_ could also function by binding and hydrolyzing GTP.

To test whether the HLP_PM19_ could bind GTP, we performed an in vitro GTP-binding assay, Trx-HLP_PM19-ΔTM-His-Strep_; AtToc33_ΔTM-R130A-His_ as positive control and Trx_His-Strep_ and BSA for negative controls were incubated with radioactive labeled [α^32^P]GTP. Aliquots of the reactions were spotted on a nitrocellulose membrane, and unbound GTP was washed off. Bound [α^32^P]GTP was detected using a phosphor imager. The results in Figure 6A showed that the purified AtToc33_ΔTM-R130A-His_ as well as Trx-HLP_PM19-ΔTM-His-Strep_ were able to bind radioactive labeled [α^32^P]GTP, whereas Trx_His-Strep_ and BSA did not ). The result clearly suggests that HLP_PM19-His-Strep_ is a GTP binder. It was further investigated whether the HLP_PM19_ could function as a GTPase by hydrolyzing GTP. In the GTPase activity assay, we used a non-radioactive approach by incubating Trx-HLP_PM19-ΔTM-His-Strep_ with GTP in the assay reaction. After running the reaction in TLC, free phosphates were visualized using a molybdate–tin(II) chloride reagent. The result in Figure 6B showed that the free phosphates could only be detected in the reaction of Trx-HLP_PM19-ΔTM-His-Strep_. This suggests that Trx-HLP_PM19-ΔTM-His-Strep_ not only binds GTP but also hydrolyzes GTP, releasing monophosphate.

### 3.8. HLP_PM19_ Functions as Nucleosidase

The results clearly showed that HLP_PM19_ binds and hydrolyzes GTP. We further questioned whether HLP_PM19_ specifically reacts with GTP or could bind and hydrolyze other types of nucleotides. To investigate whether the HLP_PM19_ could function as nucleosidase, the Trx-HLP_PM19ΔTM-His-Strep_ was incubated with purine and pyrimidine nucleotides in hydrolysis reactions. After incubation, free phosphate was detected using molybdate–tin(II) chloride reagent (Figure 7A) or in TLC analysis (Figure 7B). The result showed that Trx-HLP_PM19-ΔTM-His-Strep_ could indeed hydrolyze all nucleotides tested (Figure 7), suggesting that the HLP_PM19_ acts as a nucleosidase.

## 4. Discussion

Phytoplasmas infect various plant species worldwide, causing dramatic yield losses in crop production. DNA sequencing and genome-wide analyses show that phytoplasma genomes contain several proteins of unknown function [9]. One of them is HLPs that isfound in several phytoplasmas; however, its function is not yet known. We analyzed HLP_PM19_ to verify its localization, protein-binding, and biochemical properties. HLP_PM19_ of ’Ca. P. mali’ was identified as TlyC type, having a molecular weight of 49.5 kDa. The protein structure prediction revealed that it contains a DUF21, two transmembrane domains, a CBS region, and a transporter-associated domain CorC_HlyC. It was found that HLP_PM19_ binds to AtToc33 via DUF21-C domain in a Y2H assay. AtToc33 is one of two Toc-GTPase homologs that exist in *A. thaliana* as part of the translocon complex at the outer membrane of chloroplasts (Toc complex). It is a binding protein of precursors that is targeted to the chloroplast [14,25,26].

It is known that phytoplasma exclusively inhabit nutrient-rich phloem tissues, where they have been documented by electron microscopy [27,28]. Since phloem companion cells of a variety of plants contain chloroplasts, which represent a specific type of plastid [29,30,31] and AtToc33 was found to be localized to the plastid outer envelope [32], it is possible that the phytoplasma could attach to the chloroplast in companion cells via the interaction of HLP_PM19_ located on phytoplasma membrane with Toc33. Alternatively, since plastids and chloroplasts are derived from the same organelle, it is possible that Toc33 is also found on the sieve element plastid membrane, where the phytoplasmas have been found to localize, and HLP_PM19_ could interact with Toc33 there.

It was shown that neither nucleotide binding nor dimerization of AtToc33 is essential for chloroplast import. In addition, the absence of AtToc33 GTPase activity may be somehow compensated for by Toc159 receptors [33]. Therefore, the binding of HLP_PM19_ to the Toc33 may not strongly affect plant development, which is consistent with the finding that the transgenic plants expressing HLP_PM19_ show no phenotypic change. Although the transgenic plants expressing HLP_PM19_ did not show a severe mutated phenotype throughout plant development, it cannot be excluded that its presence and binding to Toc33 could affect the expression of the other genes. It was shown that *A. thaliana* AtToc33 knockout mutants show a severe phenotype only in the early stages of development, which gradually becomes less pronounced until it is essentially indistinguishable from Col-0 wt [34]. However, there are several genes related to photosynthetic activity that are down-regulated in these mutant plants [35]. Thus, further studies are required.

It was shown that the C-terminal part of domain of unknown function of HLP_PM19_, DUF21, is essential for the AtToc33-binding. DUFs are a large group of protein domains that are often highly conserved, which indicates an important role in biology, but remain uncharacterized [36]. Grouping proteins into a family of homologous proteins, like it is done in the Pfam database [37], can help to annotate their function. However, not all protein families have been annotated with a function so far. In 2010, in Pfam release 23.0, more than 20% of all protein domains have been annotated as DUFs [36]. DUF domains are often not essential for the organism, making them even more difficult to be characterized [36]. Therefore, the finding that the C-terminus of the DUF21 domain of HLP_PM19_ binds to AtToc33 adds a function to the DUF domain.

Phytoplasma genomes encode a variety of transporter proteins, whereas members of catabolic pathways or nucleotide production are often missing [2,3,9]. It was widely discussed that phytoplasmas take up nutrients from the host plant to survive. For ‘Ca. P. mali’, it was also hypothesized that essential compounds are received by the uptake and degradation of host plant proteins or by nucleases that produce precursors for synthesis of nucleic acids [9]. The results of the biochemical analysis of HLP_PM19_ clearly indicate that it functions as a nucleosidase, and the immuno-fluorescence histochemical staining results indicated that HLP_PM19_ is localized on the membrane of the phytoplasma, which allows the protein to bind and hydrolyze nucleotides from the host; therefore, the results presented here are the first evidence to support this hypothesis. The phytoplasma is able to multiply and circulate in the salivary glands of the host insect [38]. However, it is not yet known how the phytoplasma is able to survive in the insect. From these results, it is also possible that HLP_PM19_ functions as a nucleosidase in both the host plant and the insect for phytoplasma survival. In the host plant, it can hydrolyze nucleotides or hijack them from Toc33 while binding and possibly transporting them into the phytoplasma cells while in the host insect; further investigation is required.

## 5. Conclusions

It is not known how phytoplasmas survive in hosts, as their genomes do not encode essential proteins required for catabolic pathways or nucleotide production. Since HLP_PM19_ binds and hydrolyzes all types of nucleotides, these results strongly suggest that HLP_PM19_ functions as a nucleosidase. In addition, the results showed that HLP_PM19_ localizes on the phytoplasma membrane, allowing it to bind and hydrolyze nucleotides in the phloem, its habitat, and possibly transport them into the cells. Alternatively, it could hijack GTP from Toc33 through its binding. These findings add some light to the understanding of how the phytoplasma can survive in host cells despite the fact that its genome lacks genes encoding several functional biosynthetic and metabolic proteins.

## Figures and Tables

**Figure 2 microorganisms-13-01150-f002:**
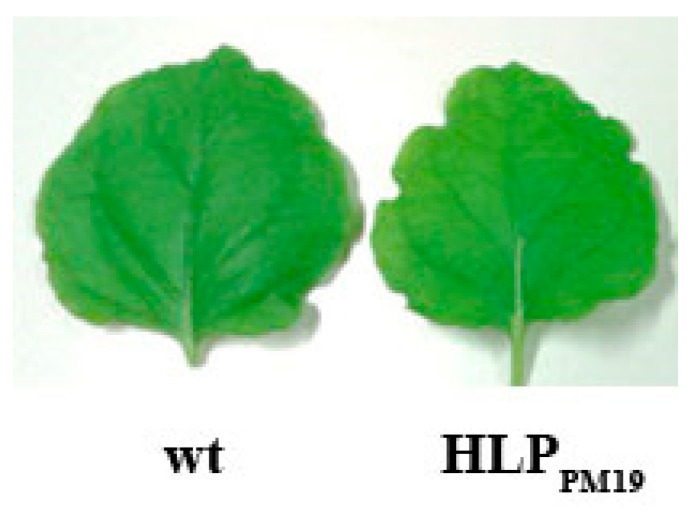
Leaves of transgenic *N. benthamiana* plants expressing HLP_PM19_ under the control of 35S promoter. No remarkably different phenotypes were observed between wild type (wt) and transgenic plant expressing HPL_PM19_ lines. The same results were also obtained from transgenic *A. thaliana* plants expressing HPL_PM19_ and HPL_PM19∆TM_ under 35S and AtSUC2 promoter.

**Figure 3 microorganisms-13-01150-f003:**
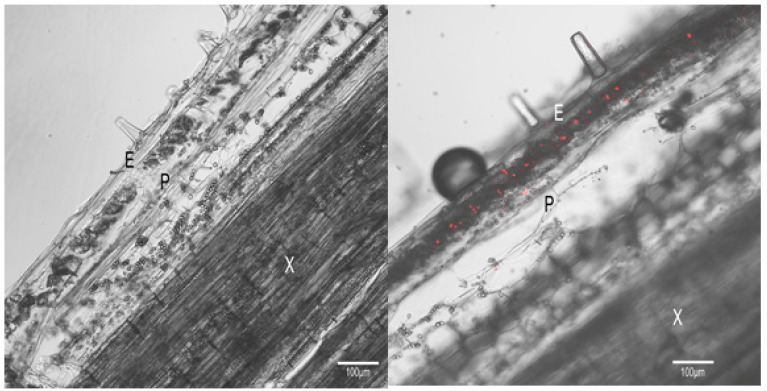
Immuno-fluorescence histochemical staining of ‘Ca. P. mali’ strain AT using anti-HLP antibody. The stems of wild type and ‘Ca. P. mali’ strain AT-infected *N. occidentalis* were sectioned longitudinally. Whole-mount immuno-fluorescence staining of ‘Ca. P. mali’-infected *N. occidentalis* at 30 days after infection using a rabbit polyclonal anti-HLP, followed by anti-rabbit-Alexa Fluor 564 (red). Epidermis (E), xylem (X), and phloem (P) are shown. Images are representative of more than three experiments. Bar is 100 µm.

**Figure 4 microorganisms-13-01150-f004:**
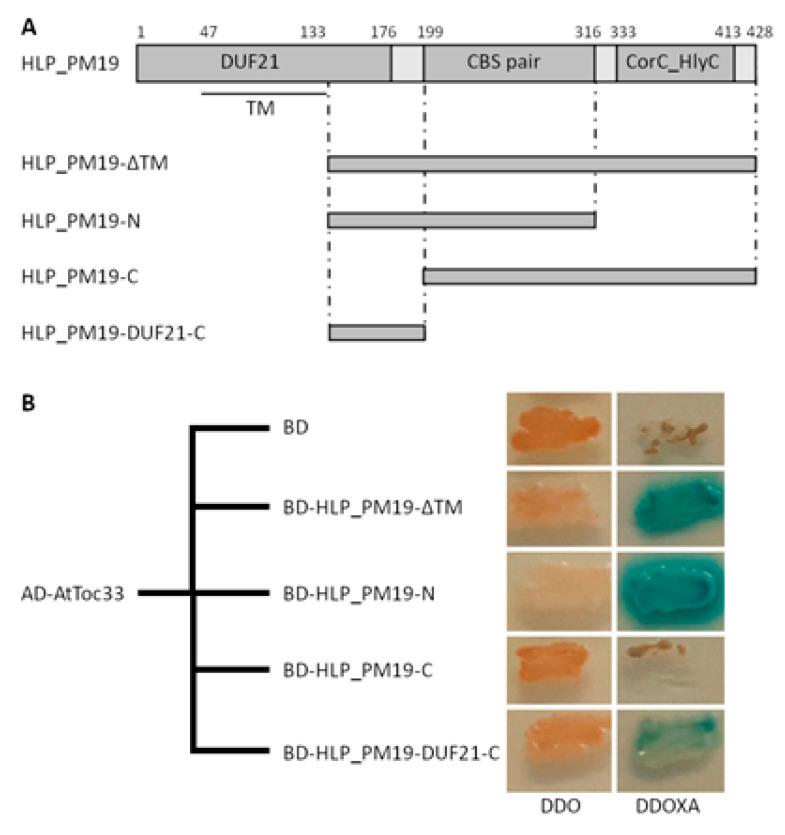
Y2H results and binding domain analysis of HLP_PM19_. (**A**) Different constructs used in Y2H screen. Domains correspond to marked annotations in Figure 1B. Transmembrane region (TM, amino acids 47 to 133) and predicted intracellular N-terminal region (amino acids 1 to 46) were deleted in all constructs (ΔTM). (**B**) Y2H results of AtToc33 with HLP_PM19-ΔTM_ and different deletion mutants. Y2H screen was performed using the Gal4 binding domain (BD) fused to different deletion mutants of HLP_PM19_ and the Gal4 activation domain (AD) fused to AtToc33, as negative control empty pGBKT7 expressing only the BD was used. Co-transformed yeast cells were patched on double drop-out medium (DDO) to select for presence of both expression plasmids and DDO media containing Aureobasidin A and X-α-Gal (DDOXA) to select for protein interaction. Growth on DDOXA and blue coloration indicate interaction of the protein partners.

**Figure 5 microorganisms-13-01150-f005:**
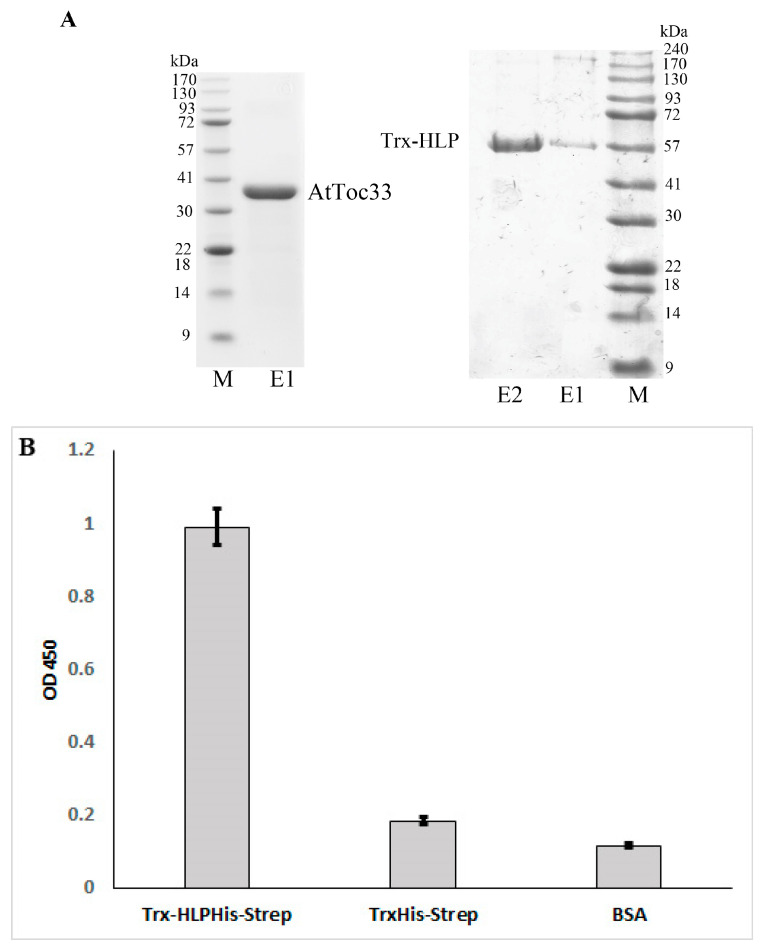
The binding of HLP_PM19_ with AtToc33 was confirmed by ELISA. (**A**) SDS-PAGE of recombinantly expressed and purified proteins AtToc33_ΔTM-R130A-His_ (AtToc33) and Trx-HLP_PM19-His-strep_. After separation of proteins using SDS-PAGE, the gel was stained with coomassie brilliant blue. M is a protein marker. E1 and E2 are protein eluents 1 and 2, respectively. (**B**) ELISA results of proteins shown in A. AtToc33_ΔTM-R130A-His_ (AtToc33) or elution buffer were used for antigen-coating. Trx-HLP_PM19-ΔTM His-Strep_ or Trx_His-Strep_ were added. Protein binding was detected using anti-strep-tag antibody followed by anti-mouse conjugated with peroxidase (POD). Colorimetric reaction was started with tetramethylbenzidine (TMB) and stopped with sulfuric acid after 2 min. Yellow coloring was measured using an ELISA reader at 450 nm. Columns show means and standard deviation of enzyme activity of three independent experiments.

**Figure 6 microorganisms-13-01150-f006:**
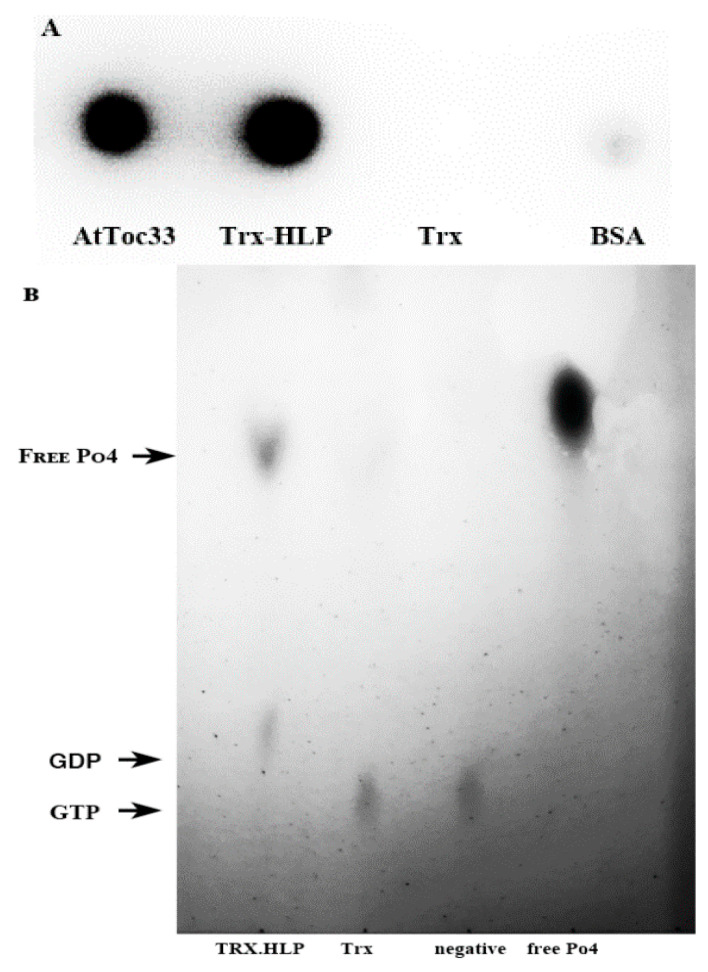
GTP-binding and hydrolysis assay with HLP_PM19_. (**A**) Recombinant AtToc33_ΔTM-R130A-His_ (AtToc33), Trx-HLP_PM19-ΔTM-His-Strep_, Trx_His-Strep_, and BSA were incubated with radioactive [α32P]GTP. Aliquots of the reaction were spotted on nitrocellulose membrane. After washing, bound [α32P]GTP was detected using a phosphor imager. The experiment was repeated three times, with similar results. AtToc33 was used as a positive control. Trx_His-Strep_ and BSA were used as negative controls. (**B**) Trx-HLP_PM19-ΔTM-His-Strep_, Trx_His-Strep_, and BSA were incubated with GTP in the hydrolysis assay reaction. After incubation, the reactions were separated on TLC. The free phosphates were visualized with molybdate–tin(II) chloride reagent. Trx_His-Strep_ and BSA were used as negative controls, and monophosphate of (1 mM potassium hydrogen phosphate) was used as a positive control. The experiment was repeated three times, with similar results.

**Figure 7 microorganisms-13-01150-f007:**
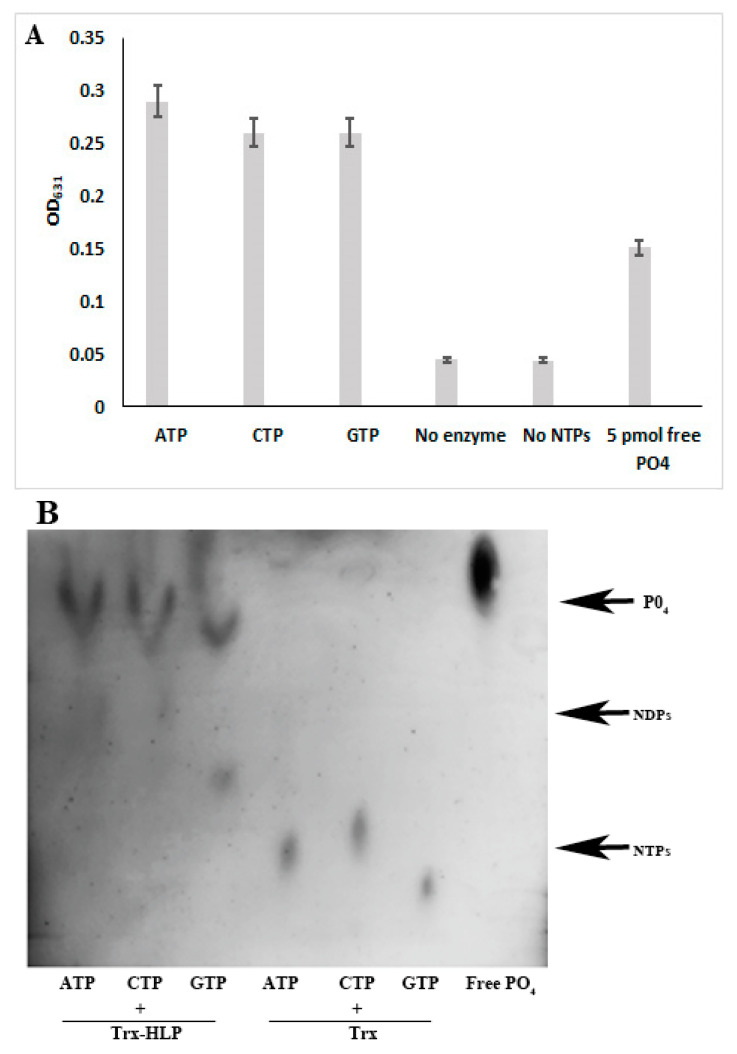
NTP hydrolysis assay. Recombinant proteins Trx-HLP_PM19-ΔTM-His-Strep_, Trx_His-Strep_ were incubated with NTPs in a hydrolysis assay. (**A**) The free phosphate was detected using a molybdate–tin(II) chloride reagent, and the solution was measured using a spectrometer with a 631 nm filter. Columns show means and standard deviations of enzyme activity of three independent experiments. (**B**) Aliquots of the reaction were spotted and run on a Polygram CEL 300 PEI/UV254 sheet. After the front reached the finish line at about 90% of the TLC sheet length, the phosphate was visualized using a molybdate–tin(II) chloride reagent. The experiment was repeated three times, with similar results. Trx_His-Strep_ was used as negative control. NDPs are nucleotide di-phosphates and NTPs are nucleotide tri-phosphates. PO4 is monophosphate. The experiment was repeated three times, which show the same results.

## Data Availability

The original contributions presented in this study are included in the article. Further inquiries can be directed to the corresponding author.

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
