# Peer review of "Hemolysin-like Protein of ‘Candidatus Phytoplasma Mali’ Is an NTPase and Binds Arabidopsis thaliana Toc33"

_microorganisms, 2025, doi:10.3390/microorganisms13051150_

Round 1
Reviewer 1 Report
Comments and Suggestions for Authors
The manuscript "Hemolysin-like protein of ‘Candidatus Phytoplasma mali’ is an NTPase and binds Arabidopsis thaliana Toc33" by Kajohn Boonrod et al. presents a report on the role of Candidatus Phytoplasma mali protein in pathogenesis. The article as a whole is systematized and valuable, but there are a number of comments on it.
1. Since ‘Candidatus Phytoplasma mali’ is a pathogen of the apple tree (Malus domestica), it is logical to expect the use of host plants or at least phylogenetically close crops to assess phenotypic changes or interactions. The introduction of the gene into Arabidopsis or Nicotiana may not reflect the true role of the protein in the natural system. This moment overrides all the advantages of the work.
2. The authors state that HLPPM19 is “not an effector” because no phenotypic changes have been observed in transgenic plants. However, the absence of visually observable phenotypes does not exclude changes at the molecular level. For example, effector expression can cause changes in the transcriptome, metabolome, or immune response. Without RNA sequencing or qPCR analysis of the expression of immune response markers, the authors' conclusion is premature and has no clear experimental justification.
3. Unfortunately, the paper provides weak statistical processing. The data must be provided in the form of averages, standard deviations (SD) or errors (SE), as well as the number of repeats (n) and the results of statistical analysis (for example, p-value or groups according to the Duncan criterion). Without this, it is impossible to judge the reliability and reproducibility of experiments, and the results must be subjected to statistical analysis.
4. The conclusion about the transport function of CorC-HlyC is based only on the assumed domain structure, without functional tests, for example, analysis of the transport of labeled substrates through the membrane or mutant forms with domain deletion. One should be careful with the wording of the conclusions.
5. The contribution of Toc33 to the pathogenesis of phytoplasma remains unclear to me. If the protein interacts with a key element of chloroplast import (Toc33), this should have functional consequences— for example, suppression of photosynthesis or immune response. However, neither the photochemical activity nor the transcription of chloroplast genes were analyzed. This makes the interpretation of the interaction incomplete.
Correcting these inaccuracies would improve the quality of the article.
Author Response
Thank you very much for your valuable comments. Below is a point-by-point response to the comments and suggestion.
- Since ‘Candidatus Phytoplasma mali’ is a pathogen of the apple tree (Malus domestica), it is logical to expect the use of host plants or at least phylogenetically close crops to assess phenotypic changes or interactions. The introduction of the gene into Arabidopsis or Nicotiana may not reflect the true role of the protein in the natural system. This moment overrides all the advantages of the work.
I agree with you. However, since we did not have a reliable mRNA apple library, we could not analyse the interaction. In addition, most of the studies on the interaction of phytoplasma proteins with plants have so far been done with Arabidopsis and Nicotiana benthamiana proteins. Furthermore, the results obtained in model plants have been used for further analysis in the actual host plant by some researchers. We therefore believe that our findings, particularly on the enzyme activity of the protein, will increase our knowledge of phytoplasma.
- The authors state that HLPPM19 is “not an effector” because no phenotypic changes have been observed in transgenic plants. However, the absence of visually observable phenotypes does not exclude changes at the molecular level. For example, effector expression can cause changes in the transcriptome, metabolome, or immune response. Without RNA sequencing or qPCR analysis of the expression of immune response markers, the authors' conclusion is premature and has no clear experimental justification.
We removed this sentence.
- Unfortunately, the paper provides weak statistical processing. The data must be provided in the form of averages, standard deviations (SD) or errors (SE), as well as the number of repeats (n) and the results of statistical analysis (for example, p-value or groups according to the Duncan criterion). Without this, it is impossible to judge the reliability and reproducibility of experiments, and the results must be subjected to statistical analysis.
We added the statistical data in our revised version.
- The conclusion about the transport function of CorC-HlyC is based only on the assumed domain structure, without functional tests, for example, analysis of the transport of labeled substrates through the membrane or mutant forms with domain deletion. One should be careful with the wording of the conclusions.
We improved wording in the revised Version of MS.
- The contribution of Toc33 to the pathogenesis of phytoplasma remains unclear to me. If the protein interacts with a key element of chloroplast import (Toc33), this should have functional consequences— for example, suppression of photosynthesis or immune response. However, neither the photochemical activity nor the transcription of chloroplast genes were analyzed. This makes the interpretation of the interaction incomplete.
Protein-protein interactions do not always change the function of the protein partner. This can be seen, for example, in Imuno dominat protein (Imp) of Ca. P. mali that bind to plant actin. The Imp binds to actin but does not change the actin filament pattern or function. It is true that the binding of hemolysin-Toc33 could have further consequences and requires further studies. We already addressed this point in the MS “Although the transgenic plants expressing HLPPM19 did not show a severe phenotype….., it cannot be excluded that its presence and binding to Toc33 could affect the expression of the other genes.”
As the funding has already ended, we have not been able to do any further experiments. For the present MS, we have added a sentence “further studies are required”.
Reviewer 2 Report
Comments and Suggestions for Authors
The authors of the manuscript microorganisms-3568262 report the protein interaction of the Hemolysin-Like Protein (HLP) of ‘Candidatus Phytoplasma Mali’ strain PM19. The authors demonstrate the HLPPM19 interact with Arabidopsis thaliana Toc33, a GTP-binding protein. The authors also present evidences that suggest a nucleoside-triphosphatase (NTPase) activity for HLPPM19. The biological significance of HLPPM19 is discussed.
The manuscript needs corrections before publication.
In the Introduction Section, the authors must include short sentence related to phytoplasma peculiarities, i.e., phloem location and transmission by phloem-feeding insects.
In the Material and Methods Section, the authors must present the experiment that were repeated three times
L64 – Italics must be used, i.e. Malus x domestica (for the scientific name).
Figure 5 and Figures 6 include graphs with error bars. The significance of error bars must be included in the Figure caption (e.g. standard deviation). Use of less dense color for graphs bars would make more obvious the error bars.
Author Response
Thank you for reviewing our manuscript and valuable comments and suggestions. Below is a point-by-point response to the comments.
The authors of the manuscript microorganisms-3568262 report the protein interaction of the Hemolysin-Like Protein (HLP) of ‘Candidatus Phytoplasma Mali’ strain PM19. The authors demonstrate the HLPPM19 interact with Arabidopsis thaliana Toc33, a GTP-binding protein. The authors also present evidences that suggest a nucleoside-triphosphatase (NTPase) activity for HLPPM19. The biological significance of HLPPM19 is discussed.
The manuscript needs corrections before publication.
1. In the Introduction Section, the authors must include short sentence related to phytoplasma peculiarities, i.e., phloem location and transmission by phloem-feeding insects.
Were added
2. In the Material and Methods Section, the authors must present the experiment that were repeated three times
Were added
3. L64 – Italics must be used, i.e. Malus x domestica (for the scientific name).
Done
4. Figure 5 and Figures 6 include graphs with error bars. The significance of error bars must be included in the Figure caption (e.g. standard deviation). Use of less dense color for graphs bars would make more obvious the error bars.
Done
Reviewer 3 Report
Comments and Suggestions for Authors
The article titled "Hemolysin-Like Protein of 'Candidatus Phytoplasma Mali' Is an NTPase and Binds Arabidopsis thaliana Toc33" may be of interest to journal readers. It also presents information relevant to the area of ​​study. However, it presents some areas that could be improved, such as: 1. It is necessary to add information on the equipment used in the methodology (specifications). 2. Correct mention of scientific names (names in italics). 3. Greater clarity and quality are suggested for the figures presented in the document. For example, Figure 1B is not very clear, so the information presented therein is not adequately appreciated. 4. It is suggested that the discussion of the results and the conclusion be expanded, given that very generalized information is provided in this last section, and the results presented in the document can support a more consistent and impactful conclusion.
Comments on the Quality of English LanguageThe document contains some grammatical and punctuation errors that need to be reviewed.
Author Response
Thank you for reviewing our manuscript and valuabel comments and suggestions. Below is a point-by-point response to your comments.
The article titled "Hemolysin-Like Protein of 'Candidatus Phytoplasma Mali' Is an NTPase and Binds Arabidopsis thaliana Toc33" may be of interest to journal readers. It also presents information relevant to the area of the study. However, it presents some areas that could be improved, such as:
- It is necessary to add information on the equipment used in the methodology (specifications).
Done
- Correct mention of scientific names (names in italics).
Done
- Greater clarity and quality are suggested for the figures presented in the document. For example, Figure 1B is not very clear, so the information presented therein is not adequately appreciated.
We improved the Fig. 1B and the original figures could be seen in the supplement files.
- It is suggested that the discussion of the results and the conclusion be expanded, given that very generalized information is provided in this last section, and the results presented in the document can support a more consistent and impactful conclusion.
Improved
Round 2
Reviewer 1 Report
Comments and Suggestions for Authors
Thanks to the authors for their efforts to improve the article.